# Scanning Ion-Conductance Microscopy for Studying Mechanical Properties of Neuronal Cells during Local Delivery of Glutamate

**DOI:** 10.3390/cells12202428

**Published:** 2023-10-11

**Authors:** Vasilii Kolmogorov, Alexander Erofeev, Alexander Vaneev, Lyubov Gorbacheva, Dmitry Kolesov, Natalia Klyachko, Yuri Korchev, Petr Gorelkin

**Affiliations:** 1Research Laboratory of Biophysics, National University of Science and Technology “MISIS”, Moscow 119049, Russia; 2Faculty of Chemistry, Lomonosov Moscow State University, Moscow 119991, Russia; 3Faculty of Biology, Lomonosov Moscow State University, Moscow 119991, Russia; 4Faculty of Biomedicine, Pirogov Russian National Research Medical University, Moscow 117997, Russia; 5Research Laboratory of SPM, Moscow Polytechnic University, Moscow 107023, Russia; 6Department of Medicine, Imperial College London, London SW7 2BX, UK; 7Nano Life Science Institute (WPI-NanoLSI), Kanazawa University, Kanazawa 920-1192, Japan

**Keywords:** scanning probe microscopy, scanning ion-conductance microscopy, cell biomechanics, nanopipette, local delivery, neuronal cells

## Abstract

Mechanical properties of neuronal cells have a key role for growth, generation of traction forces, adhesion, migration, etc. Mechanical properties are regulated by chemical signaling, neurotransmitters, and neuronal ion exchange. Disturbance of chemical signaling is accompanied by several diseases such as ischemia, trauma, and neurodegenerative diseases. It is known that the disturbance of chemical signaling, like that caused by glutamate excitotoxicity, leads to the structural reorganization of the cytoskeleton of neuronal cells and the deviation of native mechanical properties. Thus, to investigate the mechanical properties of living neuronal cells in the presence of glutamate, it is crucial to use noncontact and low-stress methods, which are the advantages of scanning ion-conductance microscopy (SICM). Moreover, a nanopipette may be used for the local delivery of small molecules as well as for a probe. In this work, SICM was used as an advanced technique for the simultaneous local delivery of glutamate and investigation of living neuronal cell morphology and mechanical behavior caused by an excitotoxic effect of glutamate.

## 1. Introduction

Cellular homeostasis, proliferation, and chemical and electrical signaling are significantly influenced by the mechanical characteristics of neuronal cells. Deviations from the original mechanical characteristics of neuronal cells can be used to diagnose several illnesses, including ischemia, trauma, and Parkinson’s and Alzheimer’s diseases, among others [1,2,3]. Excessive activation of NMDA- or AMPA-receptors and excitotoxic effects of mediators like glutamate or aspartate accompany these pathologies. It is well established that glutamate excitotoxicity causes alterations in metabolism, extracellular Ca^2+^ influx, pH, mitochondrial malfunction, and mitochondrial potential decrease [4,5,6].

Additionally, cytoskeletal proteins, which have been demonstrated to be important actors in the signal regulation and transduction of many receptors and channels, are affected by glutamate excitotoxicity [6,7,8]. For a fundamental understanding of neuronal cell death, investigation of mechanical features such as the elasticity of neuronal cells, adhesion forces, and traction stresses caused by excitotoxic effects of glutamate is crucial. Cell mechanical properties are characterized using a variety of approaches, including the use of microfluidic devices, magnetic/optical tweezers, and micropipette aspiration. However, atomic force microscopy (AFM) is frequently used for both the localized mechanical stimulation of cortical neurons as well as the research of the mechanical properties of neuronal cells [9,10,11]. AFM does have several disadvantages, such as a high applied force value.

Through noncontact scanning mode and minimal applied force, scanning ion-conductance microscopy (SICM) is more appropriate for investigating the mechanical characteristics of living neurons. In the past, SICM employed submicron pipettes to apply hydrostatic pressure through the pipette tip and deform cell surfaces; however, this method was characterized by a relatively high applied force and low resolution level because of the size of the pipettes that were being used [12,13].

Recently, we showed how nanopipettes can deform a cell via intrinsic force between the nanopipette tip and cell surface [14,15], allowing us to estimate the Young’s modulus with higher resolution and with less force applied to living cells. We demonstrated that SICM for a Young’s modulus estimate enables qualitative and quantitative characterization of the cytoskeleton state during alteration of cytoskeletal proteins such as F-actin and tubulin [16,17,18].

Also, nanopipettes in SICM can be used for the local delivery of molecules as well as for a probe. As it was reported, reagents can be loaded into a nanopipette and released due to electrophoretic flow [19,20] to living cells. Local delivery of molecules is controlled by applied hydrostatic pressure or electric potential, depending on the molecule charge. Thus, SICM allows the rapid, quantitative, and highly localized activation of specific channels and receptors of living cells. Nanopipettes were used for not only small molecules but voltage-driven DNA molecules [21], which provide new prospects for PCR, nanobiopsy, etc.

So, a combination of the local delivery of molecules, in this case the mediator, and an estimation of Young’s modulus of neuronal cells is essential for the investigation of the excitotoxic effect of glutamate on cell morphology and mechanical properties. In this work, we demonstrate a simultaneous local delivery of glutamate for the activation of glutamate receptors and an estimation of hippocampal neurons’ morphology and Young’s modulus.

## 2. Materials and Methods

### 2.1. Cell Preparation

The studies were made on primary cultures of hippocampal neurons (10 DIV) extracted from the brain of 1–3-day-old Wistar rat pups. Wistar rat pups were decapitated; hippocampi were removed from the brain and placed in Hank’s solution (Gibco, Carlsbad, CA, USA) without Ca^2+^ and Mg^2+^, containing 1 mM sodium pyruvate and 10 mM HEPES (pH 7.2), and crushed. The cells were then transferred to PBS containing DL-cysteine-HCl (0.2 mg/mL), bovine serum albumin (0.2 mg/mL), glucose (5 mg/mL), papain (0.5 mg/mL, “Sigma, Saint Louis, MO, USA”), and DNAasaI (0.01 mg/mL), and incubated at 37 °C for 5–10 min. Next, the cells were placed and dispersed in Hank’s solution (Gibco, CA, USA), without Ca^2+^ and Mg^2+^, containing DNAasaI (0.01 mg/mL) and centrifuged at 200× *g* for 4 min. The supernatant was discarded, and Hank’s solution (Gibco, Carlsbad, CA, USA), with Ca^2+^ and Mg^2+^, containing 1 mM sodium pyruvate and 10 mM HEPES (pH 7.2) was added to the precipitated cells, dispersed in saline solution, and centrifuged at 200× *g* for 4 min. The resulting pellet was resuspended in neurobasal medium for neurons (NBM, Gibco, Carlsbad, CA, USA) containing 0.5 mM L-glutamine, 2% Supplement B-27, and 100 U/mL penicillin/streptomycin (Gibco, Carlsbad, CA, USA). The resulting cell suspension (1 × 10^6^ cells/mL) (100 µL per glass) was transferred to glass bottom Petri dishes coated with PEI (1 mg/mL, Sigma, Saint Louis, MO, USA). After 1 h (37 °C, 5% CO_2_), nonadherent cells were removed, and 1.5 mL of culture medium (neurobasal medium A containing 2% supplement B-27 and 0.5 mM L-glutamine) was added. After 1 day, arabinoside (ARAC, 10^–5^ M) was added to suppress the growth of glial cells; as a result, the proportion of glial cells in the culture did not exceed 5%. Next, 1/3 of the medium in the cells was changed every 3 days. For intracellular calcium imaging, hippocampal neurons were loaded with 2 μM of Fluo-4 acetoxymethyl ester (Fluo-4 AM; Invitrogen, Carlsbad, CA, USA) for 30 min at 37 °C and 5% CO_2_. Washing out the remaining Fluo-4 was followed by 20 min for de-esterification before transferring to the perfusion chamber. Ca^2+^ fluctuations were imaged by exciting Fluo-4 at 450–480 nm. For confocal imaging of actin filaments, control neurons and neurons after glutamate application were fixed by 4% PFA in 20 min. Then, Alexa Fluor 488-Phallodidin (Invitrogen, Carlsbad, CA, USA) and Hoechst 33342 (Invitrogen, Carlsbad, CA, USA) were used for staining of actin filaments and cell nucleus. Confocal imaging was performed by using LSM 510 (Carl Zeiss, Oberkochen, Germany).

### 2.2. Scanning Ion-Conductance Microscopy

A scanning procedure was performed using SICM manufactured by ICAPPIC (ICAPPIC Ltd., Londom, UK). Nanopipettes with a typical radius of 50 nm were fabricated from borosilicate capillaries (1.2 mm, OD; 0.69 mm, ID; Sutter Instruments, Novato, CA, USA) using a Laser Puller P-2000 (Sutter Instruments, Novato, CA, USA).

Cell topography and Young’s modulus measurements were performed using the noncontact hopping mode of an adaptive resolution with maps of 40 × 40 μm, with an image resolution of 256 × 256 pixels. The approach rate during imaging was set at 100 μm s^−1^. For all SICM measurements, a three-set point mode was used. A noncontact topographic image was obtained at an ion current decrease of 0.5%, and a further two images were obtained at an ion current decrease of 1% and 2%, corresponding to membrane deformations produced by intrinsic force at each set-point [22].

Borosilicate nanopipettes loaded with Hank’s solution (HBSS) and glutamate in a final concentration of 10 mM were used for scanning topography and Young’s modulus of hippocampal neurons before glutamate delivery at −200 mV. Surface scanning by SICM allowed us to deliver glutamate and activate glutamate receptors locally at the cell soma and dendrite area. Then, a nanopipette was brought to the cell surface at a distance of a nanopipette’s radius for glutamate releases at +400 mV [19]. Successful activation of glutamate receptors and extracellular Ca^2+^ influx was accompanied by increased fluorescence intensity of Fluo-4 AM (Figure 1). Immediately following glutamate-receptor activation, glutamate release was inhibited by providing a voltage of −200 mV. Then, at intervals of 3 min, a Young’s modulus estimate and continuous topographic scanning by SICM were carried out. Nanopipettes filled with glutamic acid were used for the control measurements; however, the voltage was not switched to +400 mV. The Appendix A contains a model of concentration dependence.

## 3. Results and Discussion

In this work, we demonstrated a combination approach for the local delivery of glutamate to dendrite and soma areas, followed by measurements of Young’s modulus of hippocampal neurons. A schematic principle behind the experiments is shown in Figure 1. Increased fluorescence accompanied by Ca^2+^ influx confirms glutamate receptors. Previously, nanopipettes were used to deliver molecules locally; for NMDA receptors’ activation of thalamocortical neurons, double-barrel pipettes were used to deliver glutamate and Fluo-4 at the same time [23]. The pipettes, however, were relatively large (500–600 MΩ), whereas we used 95 MΩ nanopipettes, and local deliver was made manually by using micromanipulators without feedback control. SICM, on the other hand, enables a precise approach to the surface with nanoscale resolution. As previously reported, SICM was successfully used to activate TRPV-1 channels via voltage-driven local capsaicin application [20]. Because of the ability to obtain cell topography and select distinct locations, as well as to control the distance to cell surface with nanoscale precision, SICM is more suitable for local activation of different channels and receptors. Importantly, the final concentration of voltage-driven delivery is highly dependent on the distance between the nanopipette’s tip and the surface. The concentration of the delivered substance may decrease exponentially as the capillary is gradually removed from the surface. Because it is difficult to manage the distance between the pipette’s tip and the cell surface, manual local delivery does not provide for control of the final concentration of the delivered substance. We estimated that 3 mM glutamate (Appendix A) is required for the activation of glutamate receptors in the cell soma and dendrite. Thus, the concentration of local delivery may be controlled by the distance between the pipette’s tip and the surface.

During control measurements of topography and Young’s modulus, insignificant changes in neuron morphology and Young’s modulus were observed (Figure 2A), indicating that glutamate release is completely under control due to negative electrophoretic flow and the absence of changes in mechanical properties during cell deformation. Local glutamate delivery at the soma area increased cell volume by 40–80%, although Young’s modulus decreased considerably (Figure 2B). Local delivery to neuron dendrites resulted in a considerable decrease in Young’s modulus and a quick increase in cell volume as well. Furthermore, the small size of the nanopipette and the low applied force allowed for the observation of a decrease in Young’s modulus of neuron dendrites after glutamate-receptor activation. Statistical measurements are presented in Appendix A. We recently demonstrated [22] the application of SICM for estimating mechanical parameters of living cells based on the intrinsic force between a nanopipette’s tip and the cell surface using the Hertz model. It is known that the Hertz model has limitations when applied to a cell’s Young’s modulus estimation. The Hertz model can be used in the case of an isotropic [24] sample with a linear elastic response. Because cells do not match these criteria, measurements based on the Hertz model should be provided by low applied force and indentation depth [25] to avoid depth-dependence of mechanical properties of a cell. However, we demonstrated in our approach that the indentation value and force applied to the living cell are lower than in Peak-Force AFM. Thus, we not only match the Hertz model, but we can also determine the Young’s modulus of neuronal cells without causing mechanical activation of neurons [26].

NMDA receptors are highly connected to actin filaments, and receptor activation leads to structural reorganization of F-actin through diffusion of RhoGTPases and Ca^2+^ [6,7]. As a result, such structural alterations impair dendric spines, disrupt osmotic pressure, and reduce cytoskeleton elasticity [27,28,29]. Furthermore, activation of NMDA receptors alters cofilin regulation, which is associated with actin depolymerization factor. Correlative scanning of topography, Young’s modulus, and confocal imaging of HT-1080 cells during cytochalasin-D application revealed that estimating Young’s modulus based on the intrinsic force between the nanopipette and cell surface directly corresponds with the cytoskeleton state [22]. Thus, actin filaments of control neurons and local delivery of glutamate were stained by Alexa Fluor 488-Phallodidin for confocal microscopy. Reduced actin density following glutamate application makes structural reorganization of actin filaments clearly visible (Appendix A). Significant volume increase could be explained by changes in intracellular osmolarity caused by Na^+^ and Ca^2+^ influx.

## 4. Conclusions

We successfully demonstrated that scanning ion-conductance microscopy may be used for the local chemical simulation of living neurons as well as the investigation of mechanical properties. Nanopipettes can be used for precise, quantitative, and voltage-driven delivery of small molecules to different neuronal cell parts and activation of specific channels and receptors, like NMDA receptors. Moreover, at the same time, nanopipettes can used for the estimation of a neuronal cell’s Young’s modulus without uncontrollable release of glutamate during continuous scanning. We presume that this technique provides a new prospective for studying mechanical properties of living neural cells during excitotoxic effects of mediators.

## Figures and Tables

**Figure 1 cells-12-02428-f001:**
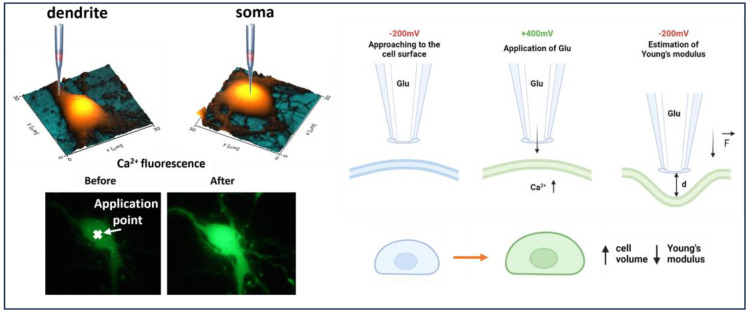
The scheme of local delivery of glutamate to neuronal cell (1-approaching to the cell surface at −200 mV, 2-applying +400 mV to release glutamate from nanopipette, 3-deformation of cell surface and estimation of Young’s modulus); increase in Fluo-4 AM fluorescence after glutamate-receptor activation and extracellular Ca^2+^ influx.

**Figure 2 cells-12-02428-f002:**
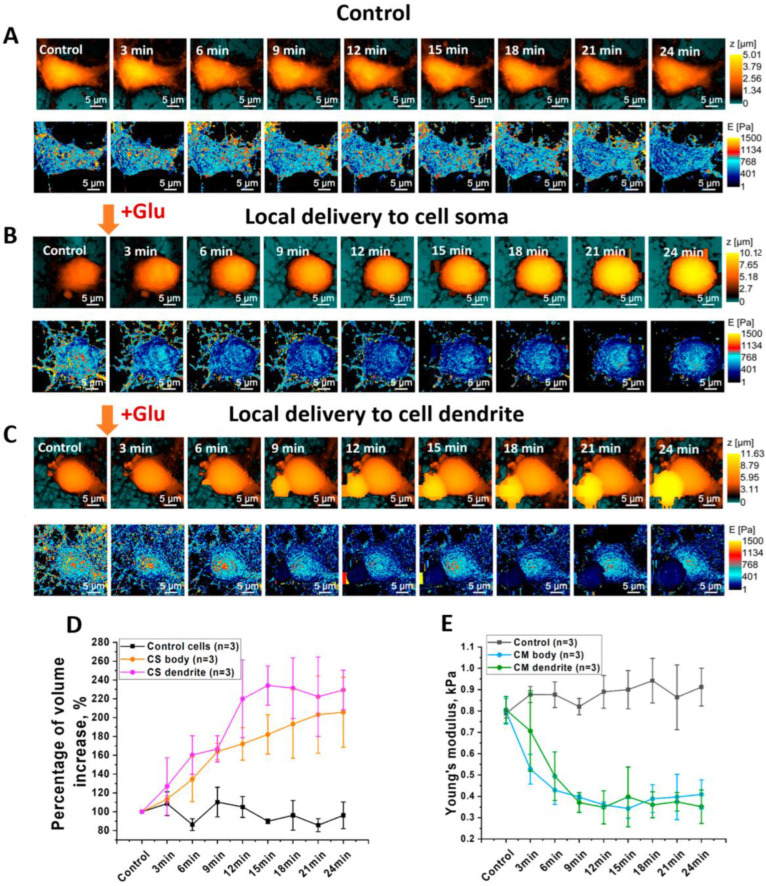
(**A**) Topography of control neuron during continuous mapping without local delivery of glutamate; (**B**) Young’s modulus distribution of control neuron during continuous mapping; (**C**) topography of neuron during continuous mapping before and after local delivery of glutamate; (**D**) mean value of cell volume increase in control cells and cells after local delivery of glutamate; (**E**) mean value of Young’s modulus before and after local delivery of glutamate (one-way ANOVA) (n—number of measured cells).

## Data Availability

Not applicable.

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
