# Peer review of "Scanning Ion-Conductance Microscopy for Studying Mechanical Properties of Neuronal Cells during Local Delivery of Glutamate"

_cells, 2023, doi:10.3390/cells12202428_

Round 1
Reviewer 1 Report
The authors report on using scanning ion conductance microscopy (SICM) for cell mechanics. The potential of using SICM is nicely described. Although I like the idea of using SICM for simultaneous induction of cell response and measurements of the outcome, my main concerns about these results are the following:
1) The Hertz model must be mentioned together with a short discussion on its applicability to this type of measurement. What are the limitations and assumptions?
2) To judge on the effect of certain compounds (here, glutamate) on cell population cannot be done on 3 cells. Statistics must be improved. I would measure at least 10 cells per case. To measure changes in Young's modulus (or cell volume) per single cell with 256 x 256 pixels takes 24 minutes (from Fig. 2). Measurements of 10 cells take 4-6 hours times 3 cases (control) means 2-3 days of the experiments. The experiments must also be repeated at least 3 times, as in all biological experiments. This defines the success of using SICM in cell mechanics induced by the delivery of various compounds to the cell interior.
3) Is it possible to develop a model describing the observed time-dependent changes?
Author Response
Review 1:
1) The Hertz model must be mentioned together with a short discussion on its applicability to this type of measurement. What are the limitations and assumptions?
A: We recently demonstrated the application of SICM for estimating mechanical parameters of living cells based on intrinsic force between nanopipette tip and cell surface using the Hertz model (Kolmogorov et al., Nanoscale, 2021). It is known that the Hertz model has limitations when applied to cell Young's modulus estimation. Hertz model can be used in case of isotropic (Kontomaris et al., Mater. Res. Express, 2020) sample with linear elastic response. Because cells do not match these criteria, measurements based on the Hertz model should be provided by low applied force and indentation depth (Lekka et al., Eur. Biophys. J., 1999; Li et al., Zhou et al., J. Mech. Behav. Biomed. Mater., 2012) to avoid depth-dependence of mechanical properties of cell. However, we demonstrated in our approach (Kolmogorov et al., Nanoscale, 2021) that the indentation value and force applied to the living cell are lower than in Peak-Force AFM. Thus, we not only match the Hertz model, but we can also determine the Young's modulus of neuronal cells without causing mechanical activation of neurons (Gaub et al, PNAS, 2019).
2) To judge on the effect of certain compounds (here, glutamate) on cell population cannot be done on 3 cells. Statistics must be improved. I would measure at least 10 cells per case. To measure changes in Young's modulus (or cell volume) per single cell with 256 x 256 pixels takes 24 minutes (from Fig. 2). Measurements of 10 cells take 4-6 hours times 3 cases (control) means 2-3 days of the experiments. The experiments must also be repeated at least 3 times, as in all biological experiments. This defines the success of using SICM in cell mechanics induced by the delivery of various compounds to the cell interior.
A: Statistical measurements are presented in Supplementary Information
3) Is it possible to develop a model describing the observed time-dependent changes?
A: Time-dependent changes neuron cell volume can be theoretically described by influx of extracellular Ca2+ and Na2+ through NMDA-receptors activated by local delivery of glutamic acid. In previous works (Sunnerberg et al., Langmuir, 2019;; Salbreux et al., Trend Cell Biol, 2012) close relationship of cell volume increase and cell stiffness reduce have been described. Young’s modulus decrease was explained by actin density decrease caused by cell volume increase (Efremov et al., Biophysical Journal, 2020). Moreover, NMDA receptors activation can also lead to to structural reorganization of F-actin through the activation of RhoGTPases and diffusion of Ca2+ (Konietzny et al., Frontiers in Cellular Neuroscience 2017). Excitotoxicity can lead to calcium dysregulation and apoptotic death of neurons, this accompanies with changes of cell’s form via reorganization of actin it has shown in other models of neuron’s damage (Kim et al., Neuropharmacology. 2002. doi: 10.1016/s0028-3908(02)00052-7; Glantz et al., Biochemistry. 2007 doi: 10.1021/bi061504y).
Reviewer 2 Report
The paper demonstrates how Scanning Ion-Conductance Microscopy can be used to chemically stimulate and at the same time analyze the mechanical properties of cells and in particular neurons. Considering their recent paper in Nanoscale, the original contribution of the work is somehow limited, but the paper is overall interesting and I would suggest accepting it.
However, I think that Authors should carefully check since I have spotted a number of broken sentences, missing verbs or simply sentences looking like a copy-paste made by mistake.
For example, in the abstract row 16 it is written “Disturbance of chemical signaling, accompanied by several diseases such as ischemia, trauma and neurodegenerative diseases”: Either the sentence is incomplete or the Authors meant something like “Disturbance of chemical signaling appears in several diseases such as ischemia, trauma and neurodegenerative diseases.”
By the same token at the beginning of the second page it is written “ .various techniques, including the use of microfluidic devices, magnetic/optical tweezers, and micropipette aspiration”, or in the same page at row 61 is written “Thus, its possibility for rapid, quantitative and highly localized stimulation of specific channels and receptors of living cells.” which looks as an incomplete sentence. Please carefully check the whole manuscript.
Some additional comments
11) In the abstract: “It’s” is correct but contracted forms are not recommended in a written text
22) On the second page is stated that “AFM does have several disadvantages, such as a high applied force value”. Of course, this could be true if AFM is operated only in contact mode since AFM can work also in contactless mode, and this can be understandable since the standard approach to measure Young’s modulus by AFM is by contact spectroscopy. However there are AFM probes with very low constant forces; while I am pretty sure that the Authors are right, I would be more specific (e.g. what is the practical minimum force I can apply by the scanning ion conductance microscopy with respect to the best AFM probes available? Or Authors could put directly a reference where the reader can find this information).
33) On the fourth page, row 150, the sentence seems to contradict what is written before, namely that it is difficult to use the distance between the pipette tip and cell surface as a control of the quantity of glutamate. Please clarify.
44) The supplementary document highlights the revision of Authors (probably it has been prepared while keeping the review mode of Word).
55) In the supplementary document it is written that “we used the formula18 to calculate the number of moles of sodium glutamate flowing out of the nanopipette:…”. Actually, I was unable to locate the formula in reference 18: Is it correct?
English is generally good, but please check the whole manuscript for broken sentences.
Author Response
Review 2:
The paper demonstrates how Scanning Ion-Conductance Microscopy can be used to chemically stimulate and at the same time analyze the mechanical properties of cells and in particular neurons. Considering their recent paper in Nanoscale, the original contribution of the work is somehow limited, but the paper is overall interesting and I would suggest accepting it.
However, I think that Authors should carefully check since I have spotted a number of broken sentences, missing verbs or simply sentences looking like a copy-paste made by mistake.
For example, in the abstract row 16 it is written “Disturbance of chemical signaling, accompanied by several diseases such as ischemia, trauma and neurodegenerative diseases”: Either the sentence is incomplete or the Authors meant something like “Disturbance of chemical signaling appears in several diseases such as ischemia, trauma and neurodegenerative diseases.”
A: We fixed the sentence
By the same token at the beginning of the second page it is written “ .various techniques, including the use of microfluidic devices, magnetic/optical tweezers, and micropipette aspiration”, or in the same page at row 61 is written “Thus, its possibility for rapid, quantitative and highly localized stimulation of specific channels and receptors of living cells.” which looks as an incomplete sentence. Please carefully check the whole manuscript.
A: We paraphrased sentences
Some additional comments
11) In the abstract: “It’s” correct but contracted forms are not recommended in a written text
A: The problem is fixed
22) On the second page is stated that “AFM does have several disadvantages, such as a high applied force value”. Of course, this could be true if AFM is operated only in contact mode since AFM can work also in contactless mode, and this can be understandable since the standard approach to measure Young’s modulus by AFM is by contact spectroscopy. However, there are AFM probes with very low constant forces; while I am pretty sure that the Authors are right, I would be more specific (e.g. what is the practical minimum force I can apply by the scanning ion conductance microscopy with respect to the best AFM probes available? Or Authors could put directly a reference where the reader can find this information).
A: We showed comparison of SICM and AFM methods for estimating the mechanical properties of living cells in our earlier work (Kolmogorov et al., Nanoscale, 2021). As previously demonstrated, SICM, which is based on the intrinsic force between the nanopipette tip and the cell surface, applies less force and causes less cell deformation than Peak-Force AFM. We put reference to our previous work where relevant data is provided.
33) On the fourth page, row 150, the sentence seems to contradict what is written before, namely that it is difficult to use the distance between the pipette tip and cell surface as a control of the quantity of glutamate. Please clarify.
A: We clarified the sentence. We meant that for local delivery of substances micromanipulators are usually used. But SICM provides nanoscale precision of control the distance between nanopipette tip and cell surface, which is crucial for control of concentration of delivered substance.
44) The supplementary document highlights the revision of Authors (probably it has been prepared while keeping the review mode of Word).
A: We fixed the problem
55) In the supplementary document it is written that “we used the formula18 to calculate the number of moles of sodium glutamate flowing out of the nanopipette:…”. Actually, I was unable to locate the formula in reference 18: Is it correct?
A: The number 18 refers to a previous publication that contained the formula. To minimize confusion, we corrected the citation. We also numbered the formulas that were presented in Supplementary Information.
Round 2
Reviewer 1 Report
The authors clarify all my questions. Minor remark all references are cited as number (number) etc.